

# Global miRNA expression is temporally correlated with acute kidney injury in mice

Rui Cui[1,*], Jia Xu[1,*], Xiao Chen[2] and Wenliang Zhu[3]

[1] Department of Nephrology, The Fourth Affiliated Hospital of Harbin Medical University, Harbin, Heilongjiang, China
[2] Department of Nephrology, Heilongjiang Province Hospital, Harbin, Heilongjiang, China
[3] Institute of Clinical Pharmacology, The Second Affiliated Hospital of Harbin Medical University, Harbin, Heilongjiang, China
[*] These authors contributed equally to this work.

## ABSTRACT

MicroRNAs (miRNAs) are negative regulators of gene expression and protein abundance. Current evidence shows an association of miRNAs with acute kidney injury (AKI) leading to substantially increased morbidity and mortality. Here, we investigated whether miRNAs are inductive regulators responsible for the pathological development of AKI. Microarray analysis was used to detect temporal changes in global miRNA expression within 48 h after AKI in mice. Results indicated that global miRNA expression gradually increased over 24 h from ischemia reperfusion injury after 24 h, and then decreased from 24 h to 48 h. A similar trend was observed for the index of tubulointerstitial injury and the level of serum creatinine, and there was a significant correlation between the level of total miRNA expression and the level of serum creatinine ($p < 0.05$). This expression-phenotype correlation was validated by quantitative reverse transcription PCR on individual miRNAs, including miR-18a, -134, -182, -210 and -214. Increased global miRNA expression may lead to widespread translational repression and reduced cellular activity. Furthermore, significant inflammatory cytokine release and peritubular capillary loss were observed, suggesting that the initiation of systematic destruction programs was due to AKI. Our findings provide new understanding of the dominant role of miRNAs in promoting the pathological development of AKI.

## INTRODUCTION

Acute kidney injury (AKI) is a common clinical syndrome mainly characterized by a rapid decline in kidney function (*Srisawat & Kellum, 2011*). The leading etiology is ischemia reperfusion injury (IRI), followed by sepsis and nephrotoxic insult (*Chertow et al., 2005*; *Mehta et al., 2007*; *Waikar, Liu & Chertow, 2008*). IRI is caused by an abrupt, transient decline in blood flow to the kidney (*Lien, Lai & Silva, 2003*) and leads to the production and release of inflammatory cytokines, infiltration of neutrophils and macrophages, and loss of peritubular capillaries (PTC; *Ishii et al., 2005*; *Furuichi, Kaneko & Wada, 2009*). It was suggested that IRI-induced PTC loss was directly associated with reduced kidney

Corresponding author
Wenliang Zhu, wenzwl@yeah.net

function and the progression to renal fibrosis (*Li et al., 2010*). Despite advances in clinical care, patients with AKI continue to have significant morbidity and mortality (*Aydin et al., 2007*). Although many therapeutic strategies have been proposed, few have been proven to effectively improve AKI, making this disease a vexing clinical problem worldwide.

MicroRNAs (miRNAs) are a superclass of endogenous small non-coding RNAs (~22 nucleotides). With the exception of miR-373 (*Place et al., 2008*), the vast majority of miRNAs can be defined as negative regulators of gene expression at the post-transcriptional level (*Bartel, 2004*). Currently, there are more than 2800 known unique mature miRNAs transcribed by the human genome (*Kozomara & Griffiths-Jones, 2014*). Approximately two-thirds of all human mRNA genes are affected by miRNA-mediated translational repression. Researchers generally agree on the high potential of miRNAs as therapeutic targets and diagnostic markers, as miRNAs are involved in nearly all human diseases, playing important roles in regulating pathologically related genes (*Asli, Pitulescu & Kessel, 2008*). miRNAs are involved in various biological processes, such as hypoxia, inflammation, cell death, and fibrosis, indicating indispensable roles in renal pathophysiology (*Saal & Harvey, 2009*; *Bhatt, Mi & Dong, 2011*; *Chung, Yu & Lan, 2013*; *Wei, Mi & Dong, 2013*; *Ma & Qu, 2013*).

Microcosmically, a wide spectrum of cellular injuries arise as AKI occurs, initiating systematic destruction programs, such as apoptotic and inflammatory pathways, and eventually reducing kidney function (*Basile, Anderson & Sutton, 2012*). In contrast, the reparative processes associated with AKI are inefficient. Thus, imbalance between injury and repair results in damaged kidney function and structure, eventually leading to renal fibrosis and progression to chronic kidney disease (*Coca, Singanamala & Parikh, 2012*). Here, we hypothesized that negative regulators of gene expression such as miRNAs may be critical repressors of cellular activities during AKI, when energy production and other cellular activities are significantly reduced. This hypothesis was supported by recent evidence that miRNAs negatively regulated the cell cycle in AKI (*Khalid et al., 2014*). Thus, miRNAs were suggested to be important initiators that contribute to the imbalance between injury and repair in AKI.

To test this hypothesis, a renal IRI mouse model was established and microarray analysis was used to detect changes in temporal expression of global miRNAs within 48 h after AKI. A potential correlation between the level of miRNA expression and kidney injury indexes (index of tubulointerstitial injury and level of serum creatinine) was tested. Additionally, temporal changes in inflammatory cytokine release and PTC loss were also determined. Together, these studies aimed to explore the role of miRNAs in promoting AKI-mediated decline in kidney function.

## MATERIALS AND METHODS

### Animals and animal models

The use of vertebrate animals was approved by the Experimental Animal Ethic Committee of the Harbin Medical University, China (Animal Experimental Ethical Inspection Protocol, No. 2009104) and followed the Guide for the Care and Use of Laboratory Animals published

by the US National Institutes of Health (8th Edition, 2011). Healthy male C57B/6 mice (20–25 g) were maintained under standard animal room conditions (temperature 21 ± 1 °C; humidity, 55–60%), with food and water, *ad libitum*, for 1 week before *in vivo* experiments. Animals were subjected to renal bilateral IRI using methods described previously (*Lin et al., 2010*). The warm ischemic time was 30 min. Sham-operated mice, which underwent the same surgical procedure without placement of the vascular clamp, served as controls (sham group). Mice were sacrificed at 6, 12, 24, and 48 h after reperfusion ($n = 8$ per time point). Plasma samples were taken from the tail vein to analyze the creatinine level using methods previously described (*Lin et al., 2010*). The mice were perfused with ice-cold normal saline via the left ventricle for 2 min, and the kidney was rapidly excised. Tissues were divided for histopathological analysis and molecular biological measurements, and portions of the kidney were fixed or snap-frozen in liquid nitrogen until analysis.

## Histological analysis and immunostaining

As described previously (*Lin et al., 2010*), a section of each kidney tissue sample was embedded in paraffin (stored at 4–8 °C), and another section was embedded in Tissue-Tek O.C.T. compound (Sakura Finetek, Torrance, CA, USA) and then stored at −80 °C. Frozen sections were obtained using a cryostat (Thermo Scientific, Cheshire, UK) at 4 μm, and paraffin sections were obtained using a microtome (Thermo Scientific, Walldorf, Germany) set at 2–3 μm. Paraffin sections were stained with hematoxylin-eosin (HE) and Masson's trichrome using standard techniques. Immunofluorescence labeling was performed using a previously described protocol (*Peng et al., 2014*). Primary antibodies against the following proteins were used for immunolabeling: rat anti-mouse F4/80 (1:200, clone BM8; eBioscience, San Diego, CA, USA), and rabbit anti-mouse CD31 (1:50, Abcam, Hong Kong). The secondary antibodies were Alexa Fluor 488-conjugated goat anti-rabbit and Alexa Fluor 488-conjugated goat anti-rat (1:200; Jackson ImmunoResearch Laboratories, West Grove, PA, USA). Nuclei were stained using 4,6-diamidino-2-phenylindole (DAPI). Images were captured using a Nikon microscope (Tokyo, Japan) and processed using NIS-Elements software (Tokyo, Japan).

All tissue sections were analyzed using 10–15 random outer medulla of kidney per mouse. The data obtained from each tissue are represented by the mean of all fields. The degree of tubulointerstitial injury in HE-stained paraffin sections and F4/80-positive cells were determined as previously described (*Cui et al., 2014*). CD31-labeled kidney sections were used to evaluate PTC loss as follows: each image was divided into 252 squares, where one square without a PTC was considered positive for loss, and the final score was presented as the percentage of positive squares.

## Enzyme-linked immunosorbent assay (ELISA)

Mouse TNF-$\alpha$, IL-1$\beta$, and IL-6 ELISA kits (R& D Systems, Minneapolis, MN, USA) were used to determine the concentrations of these markers in kidney tissue homogenates. Briefly, a total volume of 50 μL, containing 10 μL tissue homogenate and 40 μL sample diluent, was added to each sample well. Horseradish peroxidase-conjugated reagent was added to each well and incubated for 1 h at 37 °C. Tetramethylbenzidine was added after

the array was washed. The chromogenic reaction was terminated with stop solution, and the absorbance was measured at 450 nm with an iMark Microplate Reader (BIO-RAD, Tokyo, Japan). The BCA Protein Assay Kit (Beyotime Biotechnology Co., Ltd., Shanghai, China) was used to determine the total protein concentration of the kidney tissue homogenates.

## Tissue sample collection and total RNA extraction

Total RNA was extracted from 20 mg of frozen renal tissue using Trizol reagent (Invitrogen, Shanghai, China), and stored at $-80$ °C. The RNA eluate was stored at $-80$ °C. All RNA extraction samples were thawed and then restored at $-80$ °C for further experiments.

## Microarray assay

Oebiotech Technology Co, Ltd. (Shanghai, China) performed the miRNA microarray assay. Global miRNA expression was detected in the tissue samples using the Agilent mouse miRNA microarray (Release 19.0, $8 \times 60$K). Sample labeling, microarray hybridization, and array washes were performed according to manufacturer standard protocols (Agilent Technologies Inc., Santa Clara, CA, USA) as previously described (*Meng et al., 2015*). The microarray data were submitted to the Gene Expression Omnibus (GEO) database (accession number: GSE75076). For a single miRNA, miRNA abundance was defined as the raw microarray signal output normalized to the blank value 0.1. Total miRNA abundance was defined as the sum of the miRNA abundance values for all miRNAs ($n = 1,247$) detected by microarray.

## KEGG pathway analysis

Based on the microarray data, a pathway analysis on KEGG (*Kanehisa et al., 2016*) was performed to investigate whether some pathways were significantly regulated by miRNAs in mouse renal pathology. Briefly, experimentally validated miRNA-target genes in mice were retrieved from the miRNA-target interaction database miRTarbase (Release 6.0; *Chou et al., 2016*). The online tool Gene Prospector (*Yu et al., 2008*) was then used to obtain genes implicated in renal pathology by using 'kidney disease' as the search term. The bioinformatics software Cytoscape v2.8.3 (*Smoot et al., 2011*) was applied to identify associations between miRNAs and renal pathology-related mRNA genes. The DAVID functional annotation tool (*Huang da, Sherman & Lempicki, 2009*) was used to investigate whether some KEGG signaling pathways were significantly regulated by miRNAs in renal pathology when all the renal pathology-related miRNA target genes were uploaded as a gene list and the false discovery rate (FDR) was set at 0.05. Notably, only the miRNAs that were detected as presenting non-null microarray expression in least two time points were included in the analysis.

## Quantitative reverse transcription-polymerase chain reaction (qRT-PCR)

qRT-PCR was used to assess the temporal expression changes of six miRNAs including miR-18a, -34b, -134, -182, -210, and -214. Briefly, 0.4 μL total RNA was reverse transcribed into cDNA in 10 μL reactions using a high-capacity cDNA reverse transcription kit (Applied Biosystems, Foster City, CA, USA). The cDNA was diluted 100-fold and assayed by PCR
in a 10 µL reaction volume. The sequences of primers used for amplification of miRNAs and the internal control, U6, are shown in Table S1. For qRT-PCR, 2× SYBR Green PCR Master Mix (Applied Biosystems, Warrington, UK) was used according to manufacturer instructions with a Bio-Rad CFX96 Touch Real-Time PCR Detection System (Singapore). After a brief denaturation cycle (10 min at 95 °C), amplification parameters were as follows: 95 °C for 15 s and 60 °C for 1 min for 40 cycles. Cycle threshold (CT) values of miRNAs were normalized to U6, and fold change was calculated using the equation $2^{-\Delta\Delta CT}$ as previously described (*Wang et al., 2012*).

## Statistical analysis

All data are presented as the mean ± standard error of mean (SEM). Statistical analyses were performed using one-way ANOVA, Kruskal-Wallis, and Pearson correlation tests using GraphPad Prism v6.0 (GraphPad Software, Inc., La Jolla, CA, USA). A $p < 0.05$ was considered statistically significant.

# RESULTS

## AKI induced abrupt decline in kidney function

In the present study, an IRI mouse model was established by renal bilateral IRI surgery. The mouse kidneys showed substantial injury after IRI, including tubular swelling and deformation, expansion, inflammatory cell infiltration, tubular epithelial cell degeneration and necrosis, and accumulation of necrotic material within the lumen (Figs. 1A and 1B). The degree of damage increased with the time of reperfusion, which peaked at 24 h after IRI. This pathological damage of kidney tissue decreased at 48 h after IRI. Consistent with this result, both the index of tubulointerstitial injury and the level of serum creatinine were found to gradually increase from 6 h to 24 h as compared to those measured in sham animals, and then decreased slightly from 24 h to 48 h (Figs. 1C and 1D).

## Temporal miRNA expression correlates with serum creatinine level

miRNAs were previously implicated in the pathological progression of AKI (*Lee et al., 2014*). In this study, we observed for the first time a global, temporal change in miRNA expression, similar to the temporal trend of substantial injury after IRI (Figs. 1A and 2A). Furthermore, we investigated a potential correlation between global miRNA abundance and two indicators of kidney function (index of tubulointerstitial injury and level of serum creatinine). There was no significant correlation between total miRNA abundance and the index of tubulointerstitial injury ($R = 0.668$, $p = 0.22$; Fig. 2B). Comparably, our result indicated a significantly moderate correlation between total miRNA abundance and the level of serum creatinine ($R = 0.710$, $p = 0.029$; Fig. 2C).

Furthermore, six miRNAs (miR-18a, -34b, -134, -182, -210, and -214) were examined for a similar association between temporal expression and phenotype as was observed for global miRNAs (Figs. 2B and 2C). qRT-PCR was used to detect temporal changes in expression for each. As shown in Fig. 3A, all miRNAs showed a peak increase in expression at 24 h after IRI. The Pearson correlation test indicated a moderate, but insignificant degree of correlation between miRNA expression and the index of tubulointerstitial injury

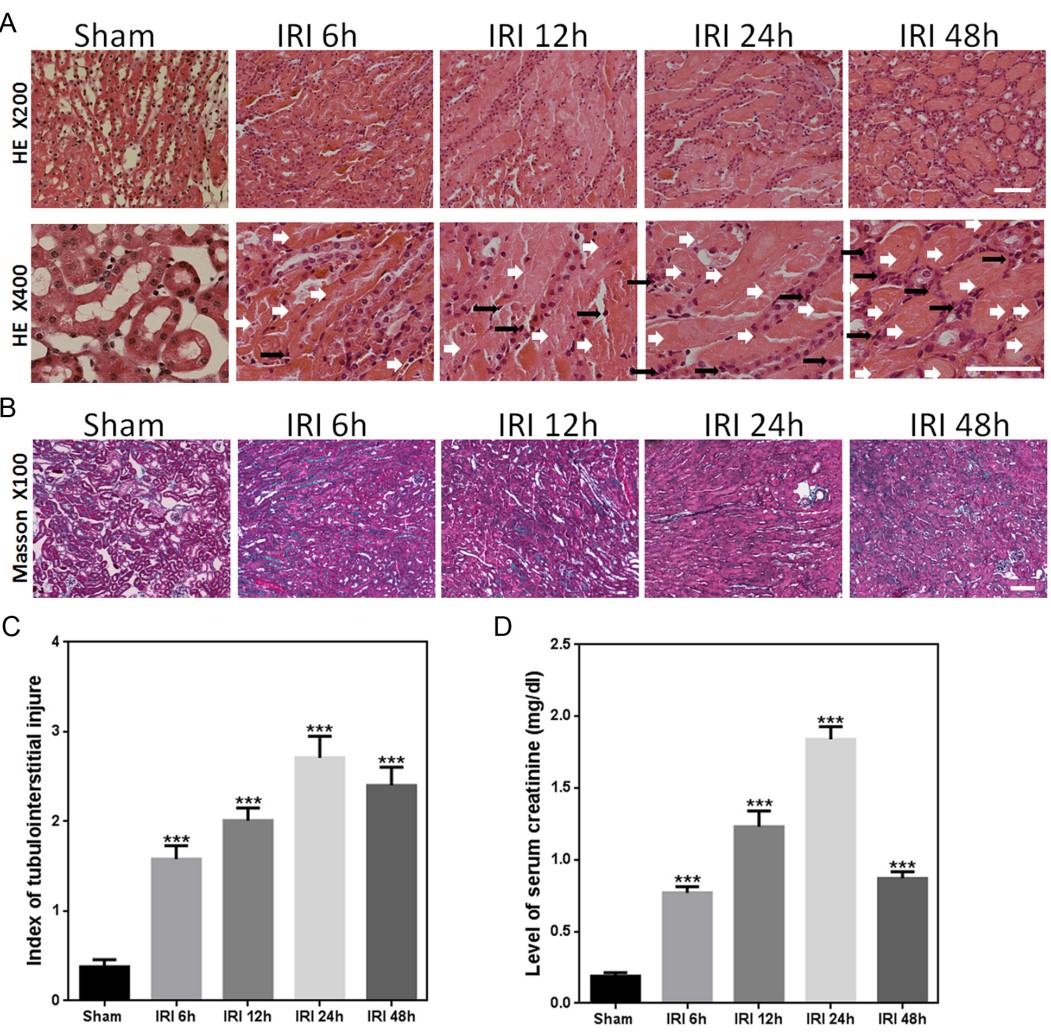

**Figure 1 IRI caused substantial changes in renal morphology and function.** (A) Representative light microscopy images of HE-stained sections of renal outer medulla from the sham and IRI groups at 6, 12, 24, and 48 h (magnification of the upper row, $200\times$ (bars = 250 $\mu$m); magnification of the lower row, $400\times$ (bars = 200 $\mu$m)). White and black arrows indicate injured tubular and infiltrated inflammatory cells, respectively. (B) Representative light microscopy images of Masson-stained sections of renal outer medulla from the sham and IRI groups at 6, 12, 24, and 48 h (magnification, $100\times$; bars = 250 $\mu$m). (C) Graph showing the score of renal tubular injury in every group. ***$p < 0.0001$ versus sham; magnification, $200\times$ $n = 8$; (D) Serum creatinine levels at 6, 12, 24, and 48 h after unilateral IRI and in the sham group. ***$p < 0.0001$ versus sham; $n = 8$.

(Fig. 3B). With the exception of miR-34b, there was a significant correlation between the expression of each of the miRNAs and the level of serum creatinine ($p < 0.05$; Fig. 3C). Notably, a correlation coefficient >0.9 indicated a strong correlation between the level of serum creatinine and temporal expression of miR-134/-214.

## miRNAs significantly regulated multiple renal pathology-related pathways

Among the miRNAs detected by microarray, 172 presented non-null microarray expression in least two time points and constructed 1,462 experimentally validated interactions with

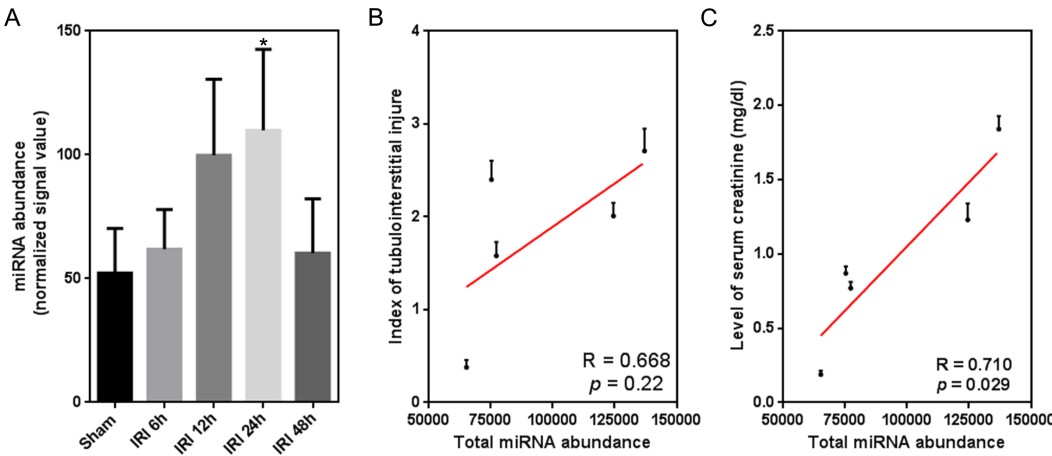

**Figure 2   Temporal correlation between the expression of global miRNAs and renal function indexes.**
(A) Temporal miRNA expression change within 48 h after renal IRI. miRNA abundance is defined as
raw microarray signal output normalized to the blank value 0.1; *$p < 0.05$ versus sham (Kruskal-Wallis
test). (B and C) Temporal correlation between total miRNA abundance and index of tubulointerstitial in-
jury/level of serum creatinine ($n = 8$). Total miRNA abundance is the sum of the abundance values of all
miRNAs detected by microarray.

548 renal pathology-related mRNA genes. A KEGG pathway analysis on the 548 target
genes revealed that miRNAs significantly regulated 10 renal pathology-related signaling
pathways (FDR < 0.05, Table S2). The Wnt signaling pathway was the top pathway that
was significantly enriched with miRNA target genes (FDR = $5.8 \times 10^{-5}$), followed by
the p53 signaling pathway (FDR = $7.6 \times 10^{-5}$) and the TGF-$\beta$ signaling pathway (FDR
= $1.0 \times 10^{-4}$). Furthermore, we investigated whether miR-18a, -134, -182, -210, and
-214 be involved in the 10 renal pathology-related pathways. Each of the five miRNAs
was experimentally validated to present kidney expression and for interactions with renal
pathology-related mRNA gene(s) (Table S3). Our pathway analysis revealed that miR-18a
targeted the *pten* gene (phosphatase and tensin homolog) in the p53 signaling pathway,
miR-210 targeted the *tcf7l2* gene (transcription factor 7-like 2) in the Wnt signaling
pathway, and the *bcl2* gene (B-cell CLL/lymphoma 2) in the apoptosis signaling pathway
(FDR = $1.2 \times 10^{-2}$). However, based on the latest release of miRTarBase (*Chou et al.,
2016*), no miRNA-target interactions were found for miR-134, -182, and -214 in the 10
pathways.

## AKI induced release of inflammatory cytokines and PTC loss
Macrophages were previously reported to be involved in both kidney injury and kidney
repair after injury (*Lee et al., 2011*). In the present study, a significant increase in the number
of macrophages was observed in IRI mice as compared to that observed in sham-operated
mice (Figs. 4A and 4B). Expression of inflammatory cytokine proteins, including tumor
necrosis factor-$\alpha$ (TNF-$\alpha$), interleukin (IL)-1$\beta$, and IL-6, significantly increased in the
kidneys of mice after IRI (Figs. 4C–4E).

  PTC loss plays a central role in kidney IRI (*Li et al., 2010*). Maintenance of the
microvasculature appears to be critical for kidney repair and preserving kidney function. To

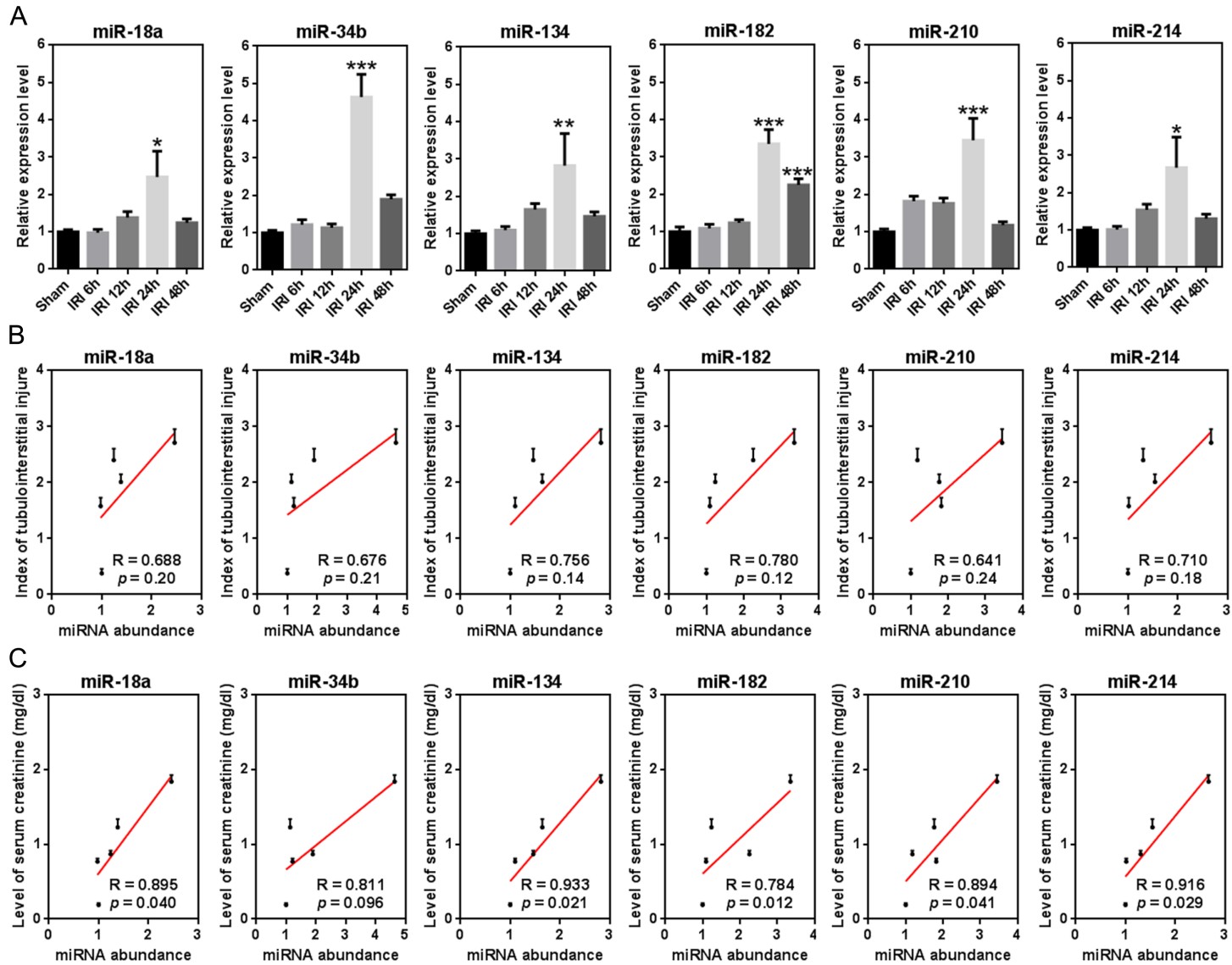

**Figure 3  Temporal correlation between the expression of single miRNAs and renal function indexes.** (A) Temporal expression changes of six miRNAs, including miR-18a, -34b, -134, -182, -210, and -214 ($n = 8$). $*p < 0.05$, $**p < 0.001$, $***p < 0.0001$ versus sham. (B and C) Temporal correlation between expression of individual miRNAs and the index of tubulointerstitial injury/level of serum creatinine ($n = 8$).

analyze peritubular capillary regeneration, we used CD31 as a marker for endothelial cells to evaluate PTC loss. PTC loss increased in kidney tissue with ischemia and reperfusion (Fig. 5A). These results are shown in Fig. 5B.

## DISCUSSION

Recently, the irreplaceable roles of miRNAs in kidney diseases, such as AKI, have been highlighted (*Bhatt, Mi & Dong, 2011*; *Wei, Mi & Dong, 2013*; *Ma & Qu, 2013*). Studies have suggested that miRNAs affect AKI pathology by interfering with regulation of the cell cycle (*Khalid et al., 2014*), and that altered miRNA profiles might be directly related

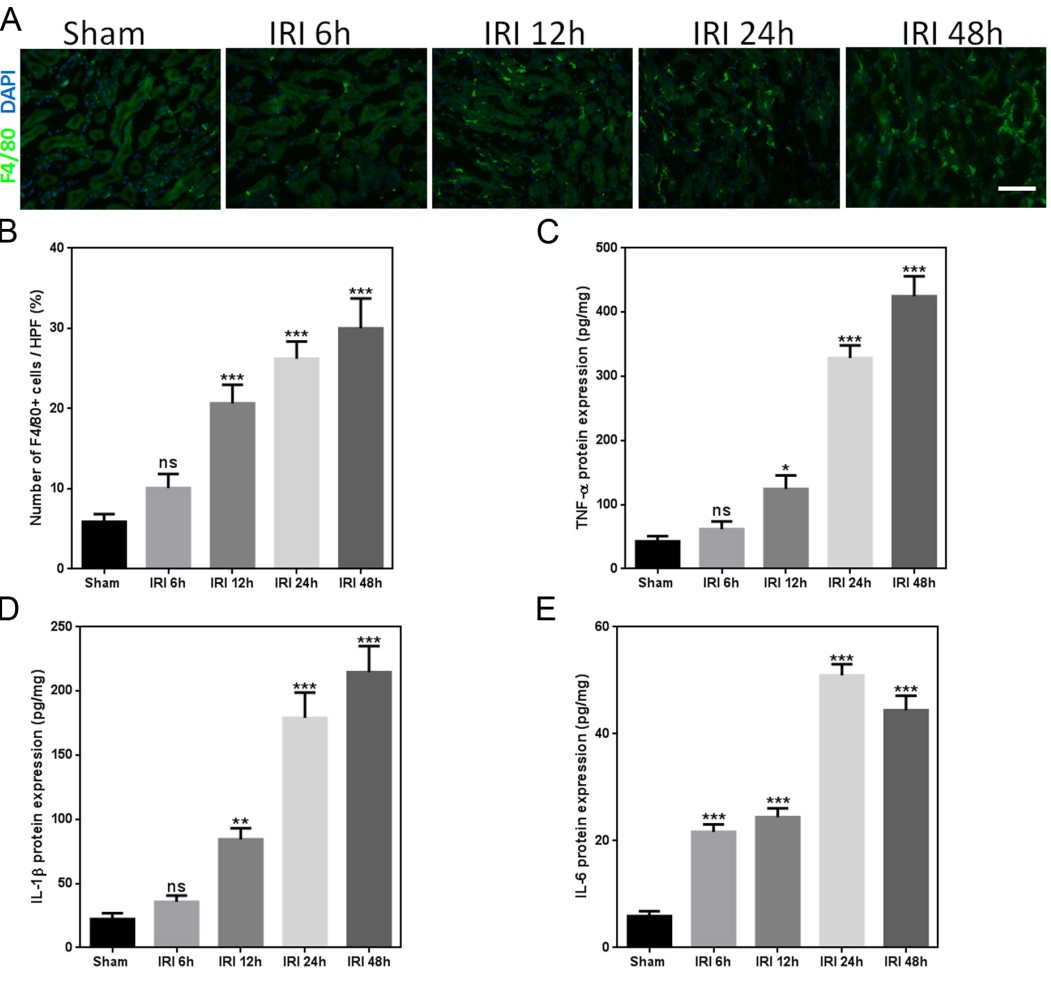

**Figure 4** **IRI induces release of inflammatory cytokines.** (A) Representative immunofluorescence images of F4/80-labeled macrophages (green) from the sham and IRI groups at 6, 12, 24, and 48 h. DAPI labels the nucleus (blue; magnification, 200×; bars = 250 μm). (B) Graph indicating the number of macrophages in mice after sham or IRI. (C–E) Representative ELISA detection results for inflammatory markers (TNF-$\alpha$, IL-1$\beta$, and IL-6) in kidney tissue homogenates of IRI mice. ns, not significant versus sham; $^*p < 0.05$, $^{**}p < 0.001$, $^{***}p < 0.0001$ versus sham; $n = 8$.

to renal dysfunction secondary to AKI (*Kumar, Liu & McMahon, 2014*). Additionally, miRNAs were identified as potential disease biomarkers for AKI (*Aguado-Fraile et al., 2013*). Therefore, current evidence strongly suggests that miRNAs may be key initiation factors in the pathological progression of AKI.

The present study investigated the temporal correlation between miRNA expression and macroscopic kidney function. Two indicators, index of tubulointerstitial injury and level of serum creatinine, were chosen as phenotypic indicators of kidney function. After establishing an IRI mouse model, the two indictors and global miRNA expression levels were measured at multiple time points within 48 h after AKI. Our results indicated abrupt aggravation of kidney injuries caused by IRI. Microarray analysis also indicated a substantial increase in global miRNA expression. This finding was consistent with a previous study by

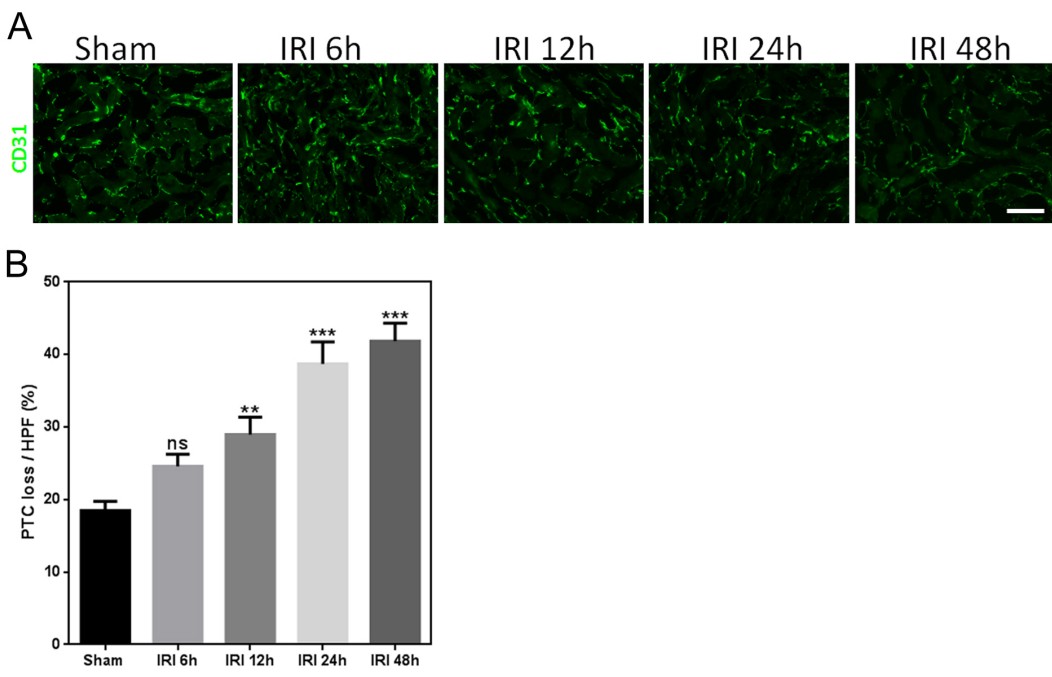

**Figure 5** **IRI causes substantial PTC loss.** (A) Representative images of mouse CD31-labeled outer medulla PTC post-ischemia/reperfusion injury kidneys at each time point (magnification, 200×; bars = 250 μm). (B) Graph showing PTC loss for mice after IRI and sham. ns, not significant versus sham; **$p < 0.001$, ***$p < 0.0001$ versus sham; $n = 8$.

*Godwin et al. (2010)*, in which the expression of many miRNAs was significantly elevated during AKI (*Godwin et al., 2010*). Our correlation analysis indicated that global, temporal miRNA expression significantly correlated with the level of serum creatinine but not the index of tubulointerstitial injury ($R = 0.71$ and $p < 0.05$). Ischemia could cause injuries to multiple sites in the kidney, such as tubules and glomerulus. It was validated that miRNAs were implicated in pathological processes in the two sites, suggesting global involvement of miRNAs in AKI (*Ho et al., 2008*; *Lu et al., 2012*). Additionally, the level of serum creatinine might more accurately reflect overall renal function than the index of tubulointerstitial injury (*Johnson & Johnson, 2005*). Based on this finding, the index of tubulointerstitial injury should be considered a local indicator of kidney function. The significant correlation between miRNA expression and a global kidney function indicator, such as serum creatinine level, is functionally equivalent to the ubiquitous roles of miRNAs in human pathophysiology (*Asli, Pitulescu & Kessel, 2008*). In particular, we demonstrated that this correlation was true for a number of individual miRNAs. The expression of five miRNAs, including miR-18a, -134, -182, -210, and -214, increased at 24 h after IRI. The significant correlation with the level of serum creatinine strongly implied an important role for the individual miRNAs in promoting the pathological development of AKI. Our result was in line with the work by *Wilflingseder et al. (2014)*, in which expression correlation analysis identified miR-182 as a key regulator of post-transplant AKI (*Wilflingseder et al., 2014*). In another clinical study, circulating miR-210 was indicated as a survival prediction biomarker for critically ill patients with AKI (*Lorenzen et al., 2011*). Until now, there was no

experimental evidence linking miR-18a, -134, and -214 to AKI. In this study, the expression of each of these miRNAs was significantly elevated. It was thought that significant expression and phenotype association provided a clue for disclosing vital miRNAs responsible for the pathological progression of AKI.

Furthermore, 10 KEGG pathways were revealed to be significantly regulated by miRNAs in renal pathology, which was consistent with previous studies reporting that miRNAs are vital regulators of signal transduction in renal pathologies (*Badal & Danesh, 2015*; *Trionfini, Benigni & Remuzzi, 2015*). Dysregulated p53 and Wnt signaling pathways were associated with the increased level of serum creatinine aroused by renal pathology (*Ghosh et al., 2015*; *Mo et al., 2015*). Our results revealed a potential role of miR-18a in regulating the p53 signaling pathway and abnormal expression of miR-210 in AKI might influence the Wnt and apoptosis signaling pathways. Our findings underscore the need for additional functional studies of miRNAs in the future.

Because they are negative post-transcriptional regulators of gene expression, increased expression of global miRNAs may result in widespread inhibition of mRNA translation into proteins. From a cellular biology perspective, an abrupt decrease in protein translation may cause reduced cellular activity and energy production (*Khalid et al., 2014*). Together, these effects would inevitably lead to structural and functional injures of the kidney at the tissue level (*Asli, Pitulescu & Kessel, 2008*). Our study revealed that global miRNA expression in kidney tissue increased after IRI. The rate of the increase in miRNA expression correlated with the rate of increase in serum creatinine level. This finding suggested a causal connection between changes in global miRNA expression and damaged kidney function in AKI. We also demonstrated continuous, steady-state release of inflammatory cytokines and PTC loss in AKI. This result may be a programmed consequence of ischemic AKI that leads to an imbalance between injury and repair (*Basile, Anderson & Sutton, 2012*; *Khalid et al., 2014*). It validated dual roles for macrophages in rapidly invading the injured renal tissue and then reestablishing a developmental program contributing to kidney repair after injury (*Lin et al., 2010*). As our experiments were performed within 48 h after acute ischemic kidney injury, the negtive effect of macrophages occupied a dominant position, explaining the further increase of F4/80 and inflammatory cytokines at 48 h.

AKI is a common medical syndrome found in hospitalized patients (*Srisawat & Kellum, 2011*). Due to a lack of effective treatment, AKI continues to be a vexing clinical problem for medical professionals worldwide. Despite recent progress in research, the pathological mechanisms underlying the AKI initiation and progression are still not fully understood (*Bonventre & Yang, 2011*; *Malek & Nematbakhsh, 2015*). Based on our observations and results from this study and a previous study (*Khalid et al., 2014*), we suggest that miRNAs act as molecular inducers of AKI progression and that altered global miRNA profiles may be responsible for pathological development of AKI. Among miRNAs, a selected few, such as miR-134 and -214, may play critical roles in this pathological course. However, further research is needed to define the pathology-initiating role of key miRNAs in AKI. In summary, our study provided novel insight into the roles of miRNAs on the progression of AKI pathology from the aspect of temporal expression of global miRNAs.

### Funding

This study was supported by research grants from the National Natural Science Foundation of China (31301095), the Funds of Health and Family Planning Commission of Heilongjiang Province (2013-110, 2014-379), and the Funds of Heilongjiang Postdoctoral Financial Assistance (LBH-Z14159, LBH-Z15165). The funders had no role in study design, data collection and analysis, decision to publish, or preparation of the manuscript.

### Grant Disclosures

The following grant information was disclosed by the authors:
National Natural Science Foundation of China: 31301095.
Funds of Health and Family Planning Commission of Heilongjiang Province: 2013-110, 2014-379.
Funds of Heilongjiang Postdoctoral Financial Assistance: LBH-Z14159, LBH-Z15165.

### Competing Interests

The authors declare there are no competing interests.

### Author Contributions

- Rui Cui performed the experiments, analyzed the data, contributed reagents/materials/-analysis tools, reviewed drafts of the paper.
- Jia Xu performed the experiments, reviewed drafts of the paper.
- Xiao Chen performed the experiments, prepared figures and/or tables, reviewed drafts of the paper.
- Wenliang Zhu conceived and designed the experiments, analyzed the data, contributed reagents/materials/analysis tools, wrote the paper, prepared figures and/or tables, reviewed drafts of the paper.

### Animal Ethics

The following information was supplied relating to ethical approvals (i.e., approving body and any reference numbers):

The use of vertebrate animals was approved by the Experimental Animal Ethic Committee of the Harbin Medical University, China (Animal Experimental Ethical Inspection Protocol, No. 2009104) and followed the Guide for the Care and Use of Laboratory Animals published by the US National Institutes of Health (8th Edition, 2011).

### Microarray Data Deposition

The following information was supplied regarding the deposition of microarray data:
GSE75076

### Data Availability

The research in this article did not generate any raw data.

# PeerJ

**Supplemental Information**

Supplemental information for this article can be found online at http://dx.doi.org/10.7717/peerj.1729#supplemental-information.

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
