# Peer review of "Global miRNA expression is temporally correlated with acute kidney injury in mice"

_PeerJ, doi:10.7717/peerj.1729_

## Round 0.1 · original submission · Major Revisions

Three reviewers have provided critical comments for this work that raise important technical issues that I feel can potentially be addressed.

Please ensure you address each point raised carefully and specifically pay attention to language and grammar, a language editing service or help from a native English speaking colleague may be suitable.

·

Basic reporting

No Comments

Experimental design

No Comments

Validity of the findings

1) Please conduct statistical test to determine significance in miRNA abundance with time (Figure 2A)
2) Based on miRNA microarray data, pathway analysis could be performed. If miRNAs belonging to a particular pathway are significantly regulated, it should be discussed and its relevance to tubulointerstitial injury or serum creatinine level could be highlighted.

Additional comments

This is a very well designed and performed study on global miRNA expression showing temporal correlation with acute kidney injury in mice. The methodology used is suitable to address the scientific questions raised by the authors. The manuscript is well written but some minor points should be addressed before publishing:

1) Authors attribute macrophages to be responsible/ involved in kidney repair after injury. Therefore, like the level of serum creatinine and miRNA abundance, there is an increase in F4/80 level and inflammatory cytokine with increasing time. But surprisingly, there is no reduction of F4/80 or inflammatory cytokines at 48 hrs? An explanation could be incorporated in the discussion.
2) Lane No 208-211 “Our correlation analysis indicated that global, temporal miRNA expression significantly correlates with the level of serum creatinine but not the index of tubulointerstitial injury” why? Possible reasons for this could be included in the discussion.

3) Lane No 145-146: Cycle threshold (CT) values of miRNAs were normalized to U6 and (fold change was) calculated using the equation 2–ΔΔCT as previously described (Wang et al., 2012).

4) The mice kidneys showing substantial injury after IRI compared to sham (tubular swelling and deformation, inflammatory cell infiltration, tubular epithelial cell degeneration and necrosis) could be highlighted in the H&E and Masson-stained sections (Figure 1A) using arrow or by adding some more figures.

Reviewer 2 ·

Basic reporting

Basic reporting of data and figures is quite acceptable. Though there are certain spelling and word choice issues like 'procession' instead of 'progression' that need to be looked at in the text.

Experimental design

The experimental design is satisfactory. The authors do not reveal the details of analysis carried out for miRNA microarray assay, though they briefly mention the calculation of miRNA abundance. Due to this, it is unclear how the choice the 6 tested miRNA was made for validation. One would expect a change in gene/miRNA expression after injury, but the authors fail to strongly define how these particular miRNAs could function in AKI. Despite the study being reasonable, it requires more work to show significance of their findings.

Validity of the findings

Though the data is robust, the reviewer would like to know how and why certain miRNAs were chosen over others.

Additional comments

None

Reviewer 3 ·

Basic reporting

No comments

Experimental design

No comment

Validity of the findings

No comments

Additional comments

1) The authors mentioned in the first sentence of the abstract that miRNA's are negative regulators of gene expression. Is that true ? For instance mRNA-373 positively regulate the gene expression of E-Cadherin (RF Place PNAS 2008).

2) Page 8 line157 why the was mentioned THE ?

3) The authors have observed several deformities in the IRI kidney tissue in figure 1. Please indicate with arrows at least for the accumulation of necrotic material in the lumen, inflammatory cell infiltration. Which is difficult to follow.

4) Explain why selecting only six miRNA's were selected to analyze in correlation to tubulointerstitial injury and creatinine levels? What are the target genes of those miRNAs and its relation to kidney IRI model ?

5) Please discuss why were the inflammatory cytokines increased in relation to increase in the miRNA's expression

---

## Round 0.2 · accepted · Accept

Thank you for your detailed responses to the reviewer comments and making suitable revisions. Congratulations!

·

Basic reporting

No comments

Experimental design

No comments

Validity of the findings

No comments

Additional comments

Authors have incorporated all the requested changes.